# Recent Advances of Triglyceride Catalytic Pyrolysis via Heterogenous Dolomite Catalyst for Upgrading Biofuel Quality: A Review

**DOI:** 10.3390/nano13131947

**Published:** 2023-06-27

**Authors:** Mohd Faiz Muaz Ahmad Zamri, Abd Halim Shamsuddin, Salmiaton Ali, Raihana Bahru, Jassinnee Milano, Sieh Kiong Tiong, Islam Md Rizwanul Fattah, Raja Mohd Hafriz Raja Shahruzzaman

**Affiliations:** 1Institute of Sustainable Energy, Universiti Tenaga Nasional, Jalan IKRAM-UNITEN, Kajang 43000, Selangor, Malaysia; 2Department of Chemical and Environmental Engineering, Faculty of Engineering, Universiti Putra Malaysia, Serdang 43400, Selangor, Malaysia; 3Institute of Microengineering and Nanoelectronics (IMEN), Universiti Kebangsaan Malaysia, Bangi 43600, Selangor, Malaysia; 4Centre for Technology in Water and Wastewater, School of Civil and Environmental Engineering, University of Technology Sydney, Ultimo, NSW 2007, Australia

**Keywords:** renewable energy, biofuel, heterogenous dolomite catalyst, thermal-activated dolomite, triglyceride feedstock

## Abstract

This review provides the recent advances in triglyceride catalytic pyrolysis using heterogeneous dolomite catalysts for upgrading biofuel quality. The production of high-quality renewable biofuels through catalytic cracking pyrolysis has gained significant attention due to their high hydrocarbon and volatile matter content. Unlike conventional applications that require high operational costs, long process times, hazardous material pollution, and enormous energy demand, catalytic cracking pyrolysis has overcome these challenges. The use of CaO, MgO, and activated dolomite catalysts has greatly improved the yield and quality of biofuel, reducing the acid value of bio-oil. Modifications of the activated dolomite surface through bifunctional acid–base properties also positively influenced bio-oil production and quality. Dolomite catalysts have been found to be effective in catalyzing the pyrolysis of triglycerides, which are a major component of vegetable oils and animal fats, to produce biofuels. Recent advances in the field include the use of modified dolomite catalysts to improve the activity and selectivity of the catalytic pyrolysis process. Moreover, there is also research enhancement of the synthesis and modification of dolomite catalysts in improving the performance of biofuel yield conversion. Interestingly, this synergy contribution has significantly improved the physicochemical properties of the catalysts such as the structure, surface area, porosity, stability, and bifunctional acid–base properties, which contribute to the catalytic reaction’s performance.

## 1. Introduction

Spurred by the efforts to mitigate greenhouse gases (GHG) emissions, liquid biofuel has received significant attention by many researchers [1,2,3]. Biofuels have played a critical role in the future of sustainable energy. As the world population continues to grow, the development of renewable energy sources, such as biofuels, provides an alternative that is sustainable, environmentally friendly, and reduces dependence on non-renewable resources [4]. The importance of biofuels and renewable energy lies in their ability to provide sustainable, environmentally friendly, and secure sources of energy to meet the growing energy demands of the future [5]. Various renewable sources and alternative feedstocks such as biomass [6,7,8], algae [9,10,11,12], cellulose biomass [13,14,15,16], and triglyceride-based agricultural fats and oils (edible and non-edible oil) have been reported producing liquid biofuels as an alternative to replace fossil fuels [17,18,19]. Biofuel is defined as liquid or gaseous fuel that can be produced from the utilization of renewable sources that offers environmental benefits [20,21]. There are several methods in producing biofuel depending on the types of renewable feedstock and the final biofuel products required. The common method of producing biofuel for the diesel range is the transesterification of triglyceride into alkyl esters, which has currently been applied in commercial biodiesel production [22,23]. However, the biodiesel produced exhibits low oxidative stability due to the tendency of vegetable oil derivatives to deteriorate through hydrolytic and oxidative reactions [24]. The biodiesel produced from this method also has comparatively low specific energy content due to the presence of oxygen and poor cold flow properties that can increase NOx emissions and create a major problem for engine performance [18,25,26]. Furthermore, this method requires a highly refined edible oil with a low free fatty acid content, and, for low value waste cooking oil, it is difficult to process by using this method [27,28]. In the petroleum refinery sector, two processes are being used to convert triglyceride into biofuel, which are hydrotreating and fluid catalytic cracking (FCC) [29,30]. The standard petroleum refinery unit is a very promising option to overcome this problem. The hydrotreatment of vegetable oil and animal fats by the petroleum refinery infrastructure produces liquid alkanes ranging from C_12_ to C_18_, which can be used as high-quality biofuel [31,32]. Unfortunately, this process requires a large amount of hydrogen to break down the double bond and to remove the oxygen present in the triglyceride molecules. In addition, it is also not economically feasible due to the additional conditioning steps in the process that can lead to higher operating and capital costs [33]. This, consequently, makes catalytic pyrolysis the most viable process to convert triglyceride to lighter and more valuable hydrocarbon products [34,35].

The catalytic pyrolysis process shows several prominent advantages in comparison with the esterification–transesterification and hydro processing of the heavy oil industries. Principally, catalytic pyrolysis uses a lower temperature range (390–450 °C) compared to conventional pyrolysis (500–850 °C) [34,36]. The biofuel produced from the esterification–transesterification process contains high oxygenated compounds that are not suitable for direct use [37]. In addition, in the hydro processing system, a large amount of hydrogen is needed to remove oxygen present in the triglyceride molecules, and this directly could increase the production cost. Therefore, catalytic pyrolysis is the most viable method to be used in biofuel production due to the quality and selectivity of products that can be improved through catalyst utilization. This innovative approach, combined with renewable fuels, can improve engine emissions and the fuel economy. In addition, as reported by previous research findings, innovative technologies could give a potential boost to the engine fuel economy and emissions reduction [38,39,40,41].

In the pyrolysis process, there are several main factors that affect the conversion, product yield, and product distribution in biofuel production. This thermochemical process is mainly influenced by feedstock composition as well as process conditions that include temperature, residence time, pressure, heating rate, and reactor design. However, the catalyst is the most significant factor to facilitate the decomposition of biomasses, promote desirable chemical reactions, and control the formation of unwanted products. The type, amount, and properties of the catalyst used can significantly affect the pyrolysis process and the resulting biofuel products. Over the past years, heterogeneous catalysts have been practically used compared to homogeneous catalysts. The homogeneous catalysts in biofuel production have several disadvantages such as the difficult and costly catalyst separation, inability to reuse and recover the catalyst, and high generation of waste and effluents [42,43]. To overcome these problems, heterogeneous catalysts have been successfully developed and utilized to cope with the environmental and economic drawbacks of the homogeneous process [44,45]. Ideally, a dolomite catalyst has high selectivity and high yield for the formation and conversion of biofuel [46,47,48]. In addition, the expensive cost of noble catalysts is not suitable for large-scale biofuel production. Thus, many novel metal–dolomite-based catalysts have been commonly used for processes due to their availability and economic feasibility [12,49,50]. Nevertheless, the discussion on the performance and efficiency of dolomite catalysts in recent catalytic processes is still very limited and has not been focused on or well discussed.

As far as has been concerned, most of the investigations of dolomite catalysts for the pyrolysis reaction have focused on the discussion on the process conditions in improving the reaction time, efficiency rate, and the yield of the biofuel produced [51]. There is very limited discussion reported on the versatility of the dolomite catalysts in upgrading the quality of the biofuel from the catalytic pyrolysis reaction. Dolomite catalysts can be easily modified or doped to further enhance their catalytic activity, stability, and selectivity. This flexibility allows for tailoring the catalyst properties to suit specific feedstocks or process conditions, providing versatility in biofuel production [52]. Furthermore, the application of raw dolomite catalysts in upgrading bio-oil having low stability and selectivity under harsh conditions has been the main concerns. These challenges can be addressed through catalyst modification, process optimization, regeneration techniques, stability-enhancing measures, and exploring alternative catalysts or production methods. The type and properties of the catalyst, such as its composition, surface area, porosity, and acidity/basicity, can all influence the mechanisms of catalytic pyrolysis of triglycerides and the formation of desired products. Different catalyst modifications may have different effects on the pyrolysis process, and their properties can be tailored to achieve specific catalytic performances and product outcomes. Therefore, this manuscript aims to provide a review on the solid alkali catalysts, named dolomite, for upgrading biofuel quality via the catalytic pyrolysis reaction. In this attempt, the latest approach of dolomite as catalysts with their advanced reaction mechanisms will be elaborated on. This review offers the recent advances of dolomite catalysts in catalytic pyrolysis reaction approaches for upgrading biofuel quality. The catalytic pyrolysis of the triglyceride feedstocks using dolomite catalysts will be explored and presented. In addition, the current progress of dolomite catalysts in catalytic reactions will be presented and elaborated on. More importantly, the key elements of dolomite catalysts for upgrading biofuel quality as well as the challenges and future directions are included and discussed. It was believed that this review could provide further research and development ideas in this area that could lead to the implementation of dolomite catalysts as an effective and sustainable catalyst option for biofuel production.

## 2. Catalysis in Catalytic Pyrolysis of Triglyceride Feedstocks

Pyrolysis is a thermal cracking process of organic materials in the absence of oxygen. It is an alternative process for producing renewable bio-based products for fuel applications. Commonly, pyrolysis of triglyceride feedstocks is applied to edible oil and fats, which produces materials from simple cracking of the primary products to various secondary products. However, the conversion and yield towards light hydrocarbons increases with the severity of the reaction conditions [53,54]. The type of feedstock and operation conditions gives the effects on the conversion, product yield, and product distribution in biofuel production using this thermal reaction. Many types of feedstocks were subjected to thermal degradation at 300–500 °C or higher and at moderate pressure [55,56,57]. Table 1 lists the relevant works dealing with catalytic pyrolysis process, indicating the present feedstock, reaction condition, catalyst, reactor, and products.

Catalytic pyrolysis involves thermal heating in the presence of a catalyst that can direct the process towards the formation of the desired products [58]. Various triglycerides such as palm and soybean oils have been highly utilized due to their mass production worldwide [59]. Most of the studies were conducted in a fixed bed reactor using zeolite-based catalysts at an ambient pressure, a temperature range of 300–500 °C, and varying hourly space velocities (WHSV). The generated products typically included organic liquid products, water, coke, and gaseous products (C_1_–C_4_ hydrocarbons, CO, and CO_2_). The organic liquid products consisted of a composition of hydrocarbons corresponding to gasoline and middle distillate (kerosene and diesel) boiling ranges. No significant amounts of oxygenated compounds were generated in the organic liquid product since the initial oxygen in the feedstock was primarily converted to water, CO, and CO_2_.

The mechanisms of catalytic pyrolysis of triglyceride have been investigated by many researchers through the utilization of different types of catalysts. Most of them agreed that the primary step involves the initial thermal decomposition of triglycerides to generate heavy oxygenated hydrocarbons by means of a free radical mechanism that is dependent on catalyst properties and characteristics. Principally, based on the main mechanism proposed by Leng et al. [60], the oils feedstock is thermally decomposed before producing heavy hydrocarbons, due to the presence of the cracking catalyst.

**Table 1 nanomaterials-13-01947-t001:** Feedstock, reaction characteristics and products of catalytic pyrolysis research.

Feedstock	Condition	Catalyst	Reactor	Product/Yield	References
Sunflower Oil	330–380 °C WHSV: 0.66/h	Vanadium pentoxide/1–2 wt%	Fixed-fluidized bed	Hydrocarbon (89.6–90.2 wt%) Gas (9.2–11.2 wt%) Coke (2–4 wt%)	[61]
Palm Oil	450 °C WHSV: 1.6–4.58/h	Na_2_CO_3_	Stirred sludge bed reactor	Hydrocarbon (61–88 wt%) Oxygenates (11.9–38.9 wt%) Coke (2–4 wt%)	[62]
Canola Oil	375–500 °C WHSV: 2–4/h	HZSM-5	Fixed bed tubular reactor	Hydrocarbon (74.9–93.6 wt%) Aromatic (0.66 to 1.7 wt%)	[63]
Woody Oil	480 °C Catalyst: 5% Flow rate: 50 g/h	CaO	Glass Vessel Reactor	Gasoline, Kerosene, Diesel (77%)	[64]
Soybean Oil	500 °C WHSV: 0.01/h	ZSM-5	Fixed bed reactor	Hydrocarbon (3–54%) Coke (30%)	[65]
Waste Cooking Oil	400 °C 1.5 h Heating rate: 20 °C/min	K_2_O/Ba-MCM-41	Pyrolysis Reactor	Cracking oil	[66]
Triolein	380 °C 0.5 g catalyst 2 h	Ni/HMS	Glass made reactor	95% of diesel range (C_11_–C_20_)	[67]
Used Vegetable Oil (UVO)	400 °C Catalyst: 1 g 3 h WHSV: 4/h	Ni, Mo, NiMo functionalized zeolite	Fixed bed reactor	Conversion to hydrocarbons: palmitic acid (84.72%) and stearic acid (74.10%)	[68]
Waste Cooking Oil	400–600 °C 2 g catalyst	Ni/Corn Activated Carbon (AC)	Fixed bed reactor	Hydrocarbons (68.47–86.38%) Syngas (CO and H_2_) (70%)	[69]
Jatropha Oil	600 °C	Ni2P/Zr-SBA-15	High-pressure fixed bed reactor	C_3_ to C_14_ (52.92 wt%) C_15_ (12.93 wt%) C_16_ (4.56 wt%) C_17_ (22.27 wt%) C_18_ (3.01 wt%) C_19_ to C_25_(4.32 wt%)	[70]

As shown in Figure 1, the heavy oxygenated compounds produced from the triglycerides thermal catalytic process undergo various reactions based on the formation of different types of hydrocarbons involving cracking, deoxygenation, dehydration, and aromatization with the H-transfer reaction. The reaction of cracking and condensation happens simultaneously and converts the triglycerides to the refined hydrocarbons. In the process, the heavy hydrocarbons and light alkanes with water, carbon dioxide, and carbon monoxide are produced from the deoxygenation reaction and cracking process, respectively [60]. Particularly, the pyrolysis compounds produced from the process are determinedly reacted based on the pyrolysis conditions such as pyrolysis temperature, heating rate, residence time, and reaction time. Based on previous studies, the reaction pathways of triglyceride are mainly converted into heavy oxygenated compounds through radical formation that is externally enhanced by the catalyst’s presence at a temperature between 330 °C and 600 °C [61,63,69,70]. Figure 2 shows the formation of fatty acids that are catalytically converted into various products such as light heavy cycle oil (LHCO), light cycle oil (LCO), gasoline and gas while entering the pores of catalyst.

Nevertheless, the insight of the reaction pathways for triglyceride transformations are different, based on the heavy oxygenated compounds’ catalytic cracking mechanisms. Moreover, the route of the intermediate compounds varies based on catalyst. In addition, the high amount of aromatic formations predominately favors the LCO and LHCO boiling ranges that led the competitions among the triglyceride transformations under different catalytic cracking mechanisms [71]. The routes of the catalytic cracking mechanisms for heavy oxygenated compounds produce various classes of hydrocarbons, paraffinic fragments, and lower aromatics [72]. Subsequently, the aromatization process of the remaining lower aromatic hydrocarbons primarily converts the saturated hydrocarbons in the gasoline range or LCO range. In an ideal process, non-saturated molecules are not wanted, due to the reactions with the impurities or within the molecules in a polymerization reaction to form unwanted big molecules [73]. However, there are several oil-based feedstocks that have the structures of primary acids with unsaturated sites that can reduce the number of reactive free radicals that produce open short unsaturated and saturated hydrocarbons with low carbon contents [74].

Predominately, zeolite-based catalysts are mostly selected in the catalytic cracking of triglycerides due to their attractive properties such as their high hydrothermal stability, strong acidity, and shape selectivity. Wang et al. [34] reviewed the various zeolites based on catalysts for the enhancement of hydrocarbon fuels to achieve large-scale production from waste oil as the feedstock. Modified zeolite catalysts are popular to promote the cracking of pyrolysis vapors because of their molecular structures and strong acid sites, and the yield of bio-oil can reach up to 40 wt% [75]. However, the products generated contain high acid values that affect the oxidative stability and cold flow properties due to the high oxygenated compounds present [34]. In addition, the carbonaceous coke that blocked the active sites of the compound rapidly deactivated the catalytic reaction and increased the water component that was produced along with the bio-oil from the dehydration reactions [76].

Meanwhile, metal-based catalysts also have excellent catalytic activity in building up the effective cracking catalysts for the generation of fuel oil and hydrocarbon products from triglyceride. Transition metals such as nickel- and cobalt-based catalysts have been proven to be very active and have widely been used for hydro processing in the heavy oil industry, as well as in the production of liquid biofuel [77,78]. Previous research showed that an alumina supported NiMo catalyst could be used to convert vegetable oil to alkanes and alkylbenzene [79]. In addition to that, alumina supported NiMo and CoMo catalysts are applicable as an effective deoxygenation catalyst to produce hydrocarbon products [80]. Nevertheless, the metal-based catalyst also resulted in the decrement of yield due to coke formation and high acidity. Moreover, the performance of the conversion mainly relied on the uniformity of the catalyst’s parameters [81]. The presence of high acidity sites resulted in extensive deactivation due to catalyst coking and tar formation [77].

Interestingly, the dolomite catalyst has succeeded in increasing conversion yield, reducing sintering resistance. The dolomite catalyst has widely been used for biofuel conversion in triglyceride catalytic pyrolysis. Previous research findings demonstrated that the conversion of biofuel produced using dolomite catalysts in a lab-scale fluidized bed reactor has significantly improved. Valle et al. [64] reported that basic compounds in dolomite were identified to promote the high fraction conversion of biofuel produced by upgrading the composition of bio-oil through the removal of carboxylic acids, acetol, and anhydrosugars and greatly increased the formation of linear ketone (acetone and 2-butanone) and cyclopentanones. Furthermore, as reported by Hafriz et al. [52], the dolomite catalyst is also effective in improving its catalytic activity via the deoxygenation reaction of bio-oil conversion. Excitingly, there are also other recent studies having similar findings proving the effectiveness of dolomite catalysts for biofuel conversion. As reported, the dolomite compound has been indicated as a strong base compound that affects CO_2_ adsorption effectiveness, hindering the carbon formation on the catalyst’s surface [82]. The high surface area of the activated dolomite catalyst enhanced the conversion of the high molecular weight hydrocarbons into light hydrocarbons [83]. Significantly, the dolomite catalyst has exhibited an important reactant’s role for absorbing and catalyzing the conditions of thermal degradation for the employed bio-oil components. Figure 3 shows the various types of catalyst groups in catalytic pyrolysis for biofuel production.

## 3. Dolomite in Catalytic Pyrolysis

Dolomite is a naturally abundant type of rock that is chemically known as Calcium Magnesium Carbonate Ca Mg (CO_3_)_2,_ which is an alkaline earth oxide with high basicity, low cost, less toxicity, and is environmentally friendly [84]. It is a non-toxic base catalyst that mainly consists of calcium carbonate (CaCO_3_), magnesium carbonate (MgCO_3_) with a small number of other compounds such as silica oxide (SiO_2_), iron oxide (Fe_2_O_3_), and aluminum oxide (Al_2_O_3_) [51]. Figure 4 and Table 2andTable 3 show the chemical structures, chemical properties, and the compositions of natural dolomite catalysts from different countries, respectively.

### 3.1. Raw Dolomite Catalyst

Predominantly, the chemical composition of a dolomite catalyst can have a significant impact on its catalytic activity in bio-oil upgrading, influencing the acid–base properties, surface areas, porosities, redox properties, stabilities, and deactivation behaviors. Optimizing the chemical composition of a dolomite catalyst can be an important factor in enhancing its catalytic activity and performance in the bio-oil upgrading processes. The basic nature of the dolomite catalyst mainly contributes to the catalytic activity of cracking reactions due to the presence of CaO/MgO (basic oxides) and Ca(OH)_2_ (strong basic) [87,88]. As formerly reported, the acid conversion pathway was established by thermal–catalytic cracking and the neutralization reactions with CaO, which formed calcium carboxylate and H_2_O [89]. The reaction dolomite catalyst for bio-oil upgrading was contributed to by the CaCO_3_ and MgO elements, which enhanced the reaction heat that raised the temperature. However, the raw dolomite catalyst had less basicity properties that were easily deactivated. In addition, the bio-oil produced from the raw dolomite catalytic reactions had high acidity and required an additional upgrading process in reducing the acid value. As reported by Buyang et al. [90], the quality of bio-oil using raw dolomite catalysts was further upgraded via a facile esterification reaction with methanol to reduce the acid content from 55.47% to 3.32%. Alternatively, the basicity of dolomite was increased by introducing the thermal-activated process of calcination that decomposed into CaO and MgO, which chemically had high basicity [83]. Moreover, further modification of raw dolomite catalysts has been introduced and enhanced the pyrolysis oil conversion into high quality liquid biofuels. The dolomitic materials’ enhancements for catalytic pyrolysis reactions were involved with various modifications to improve both the quality and quantity of bio-oil through the deoxygenation, decarbonylation, and decarboxylation reactions. Metal doping, thermal activation (calcination), and combined catalytic reactions are among the prominent approaches used to enhance the properties of raw dolomite catalysts in catalytic pyrolysis reactions. Table 4 shows the recent work of dolomite catalyst for catalytic pyrolysis reactions.

### 3.2. Thermal-Activated (Calcined) Dolomite Catalysts

Similar to other natural catalyst sources of CaCO_3_, the active phase of dolomite is obtained by the calcination of dolomite into CaO and MgO [96]. Equations (1)–(3) show the calcination reaction of the dolomite catalyst.
(1)$$\mathrm{CaMg}({\mathrm{CH}}_{3}{\mathrm{CO}}_{2}{)}_{4}\;\overset{380–400\;{}^{\circ}C}{\to }\;{\mathrm{CaCO}}_{3}+{\mathrm{MgCO}}_{3}+2C_{3}H_{6}O$$
(2)$${\mathrm{MgCO}}_{3}\;\overset{540\;{}^{\circ}C}{\to }\;\mathrm{MgO}+{\mathrm{CO}}_{2}$$
(3)$${\mathrm{CaCO}}_{3}\;\overset{>850\;{}^{\circ}C}{\to }\mathrm{CaO}+{\mathrm{CO}}_{2}$$

Upon calcination at temperatures >850 °C, the carbonate groups of the dolomites are decomposed and generate highly basic CaO and MgO [97]. The calcined dolomite will be in an oxide form, and its activity will decrease if carbonate is present. In the studies of dolomite catalyst enhancements reported by Hoang et al. [93], the thermal activation process improved the catalyst properties of natural dolomite. As shown in Figure 5, the raw natural consists of CaCO_3_ and MgCO_3_, while the thermal-activated dolomite (900 °C) shows XRD patterns of MgO, CaO, Ca (OH)_2_, CaMg(CO_3_)_2_, and CaCO_3_ that further enhances the catalytic pyrolysis process of bio-oil.

As reported by Asikin et al. [98], the CaO catalyst could absorb more CO_2_ either in the gas phase or liquid phase, which simultaneously removed the oxygen molecule via the decarboxylation and decarbonylation mechanism. In addition, CaO is a potential alternative for the deoxygenation catalyst for biofuel production via the cracking–decarboxylation–decarbonylation pathways. Kesica et al. [99] reported that the CaO catalyst was better than the ZnO catalyst because it could effectively remove acids from the bio-oil better than the other catalysts. In addition, CaO has a relatively high thermal stability, which makes it suitable for use in high-temperature pyrolysis or hydrodeoxygenation (HDO processes). It can maintain its catalytic activity at elevated temperatures, allowing for efficient deoxygenation reactions during biofuel production compared to ZnO catalysts [100]. In addition, other studies of sugarcane oil (Saccharum officinarum L) against thermal-activated dolomite catalyst have greatly enhanced the catalyst properties compared to raw natural dolomite. As reported by Charusiri et al. [94], the activated dolomite had established a pore structure with uniform pores on the surface that were indicated by a total pore volume of 0.10 cm^3^/g and a surface area of SBET = 18.21 m^2^/g. These calcined dolomite catalyst surface areas and pore volume values were higher than those of natural dolomite. In addition, the catalytic reaction upgraded bio-oil with a lower oxygen content, higher gross calorific value, and decreased acid corrosion. Interestingly, the impact of thermal activation on dolomite properties significantly improved the conversion performance of upgraded bio-oil at a low pyrolysis temperature. As reported by Valle et al. [92], the catalytic behavior of activated dolomite (800 °C) was significantly affected by a sudden increment of temperature in the early stages of the reaction. The growth of Ca(OH)_2_ at low temperatures by CaO hydration proved the important role of the high basicity of Ca(OH)_2_ in the dolomite catalyst for bio-oil oxygenate conversions, which had further enhanced CO_2_ capture. In this study, 48% of feedstock has been converted into a high-quality biofuel of linear ketones (37%) and cyclo pentanones (30%) by a sudden 50 °C early in the experiment temperature (400 °C). As shown in Figure 6, solids and liquids are the main products at 400 °C as compared to 500 °C.

Meanwhile, Raja et al. [101], reported that MgO played an important role in increasing oil quality by reducing oxygen levels and converting almost all the long chain alkanes and alkenes to low molecular weight hydrocarbons in the diesel range. The potential of MgO as a based catalyst was proved by Tani et al. [102], due to its significant role in increasing the oil quality by reducing oxygen levels and directly reducing the acid values and iodine values of the cracking oil, which was suitable to be used in diesel engine vehicles. Moreover, based on Diwald et al. [103], the strength of the basic sites of the alkaline earth metal oxides studied in temperature-programmed desorption (TPD) of CO_2_ increased in the order of MgO < CaO < SrO < BaO. In addition, the MgO ions were reported among several basic oxides (such as La_2_O_3_, CeO_2_, or ZrO_2_), which were successful in minimizing and neutralizing the acidic natures of the supports [52]. Furthermore, previous findings by Kuchonthara et al. [104] proved successful in introducing alkalines (K, Ca, Mg) and carbonates (K_2_CO_3_) to enable the increments in conversion yields and the reduction in coke formations in the reactions. Consequently, this further strengthened the performance of the thermal-activated dolomites in the catalytic pyrolysis reaction. This previously shown evidence clearly proved that the catalytic behaviors of activated dolomite were significant in improving the performance of the catalytic pyrolysis reactions.

### 3.3. Metal-Activated Dolomite Catalysts

The utilization of metal-activated dolomite catalysts has been given extensive attention for improving the catalyst’s effectiveness and stability. As reported by Hafriz et al. [95], the modification of dolomite using various transition metals (Ni, Mo, Co, Mg) was usually practiced to further enrich the catalyst. Moreover, the addition of metal oxides to basic catalysts has further increased bio-oil yield, and it also minimized and neutralized the acidity of the catalytic reaction. In previous findings, the comparison between using activated dolomite and NiO-activated dolomite catalysts showed that the addition of NiO succeeded in reducing coke formation by about 50% and increased the yield of deoxygenated oil to 47%, showing resistance to sintering in Ni-based catalysts [52]. In addition, the application of low-cost transition metal oxides has successfully proven to enhance catalytic activity as an alternative to expensive noble metals (e.g., Pt and Pd) [105].

In recent findings, the attachment of Ni metal on dolomite catalyst surfaces exhibited the highest conversion compared to other metals. Based on research findings by Hafriz et al. [77], the Ni/CMD900 catalyst exhibited the highest conversion (67.0%) and high selectivity (80.2%) compared to the others (Fe, Zn, Cu, Co/CMD900), with a high proportion of saturated linear hydrocarbons corresponding to green diesel. As shown in Figure 7, Ni/CMD900 depicted the highest composition of biofuel in the produced pyrolysis oil compared to the other transition-metal-activated dolomite catalysts. Predominately, the large density of the basic site Ni-promoted catalyst (4.40 × 10^21^ atom/g) compared to the other doped metal-activated dolomite catalysts improved catalyst stability in the reaction [77]. This finding successfully proved the significant role of catalyst surface modifications affecting catalytic behaviors.

Meanwhile, the current progress of the metal-activated dolomite catalyst has been further explored for light biofuel conversion through additions at various loading amounts of Magnesium Carbonate (MgCO_3_) with the activated dolomite catalyst. As reported by Kanchanatip et al. [82], the light biofuel yields increased as the Mg/activated dolomite catalyst resulted the highest bio-oil production (84 vol%), with 65 vol% of light biofuel. The activated dolomite catalyst that was mixed with various loading contents (0, 10, 20, and 30 wt%) of MgCO_3_ underwent pyrolytic catalysis cracking at various reaction temperatures (450–550 °C).

As shown in Figure 8, the loading amount of 20 wt% Mg/dolomite catalyst at a 500 °C reaction temperature produced the highest pyrolytic biofuel yields compared to the others. Primarily, the reaction behaviors through the combination of catalyst loading and reaction temperatures successfully produced light biofuel at the highest yield. As reported, the adequate high temperature of the reaction enabled the bio-oil to be completely cracked into the range of light biofuels [106]. In addition, the sufficient loading amounts of the Mg catalyst played an important role in avoiding the secondary cracking of the pyrolytic products that facilitated the reduction in biofuel yield and the increment in the bottom product [107].

In other side, the method of metal doping on the activated dolomite catalyst also had significant effects on the biofuel conversion. As reported by Hafriz et al. [52], the precipitation technique converted a high yield of biofuel (36.4%) with less coke formation (32.0%) compared to the co-precipitation and impregnation technique for Ni metal doping into the activated dolomite catalyst. This finding reported that the precipitation of the Ni alteration to the macrostructure of the synthesized catalysts increased the possibility of coke being deposited on the inner bulk and the surfaces of the catalysts, which reduced the catalytic activity of the catalysts. However, although this finding reported the success of Ni addition by precipitation with activated dolomite catalysts, the final acidity values of the pyrolytic oil was also still higher (47 mg KOH/g) compared to the ASTM standard (<0.01 mg KOH/g). Consequently, the blending of the light biofuel was proposed to reduce the acidity before being utilized. Considering the effect of the acidity, an updated approach was required to further improve the main drawback issues of the metal-activated catalysts hindering the utilization of the produced biofuel. Table 5 shows the quality comparison between the biofuel produced with the ASTM standard.

## 4. Dolomite Catalyst Key Elements for Upgrading Biofuel Quality

The triglyceride bio-oil conversion into upgraded biofuel is mainly influenced by the condition of the catalyst’s behaviors and properties. The dolomite catalyst has important key elements that are able to enhance the quality of the pyrolytic biofuel. These elements have a significant role in enriching the catalyst behaviors and characteristics that provide perfect reactions, hindering the occurrences of reaction drawbacks. At present, the preparation methods of the porous, high surface and stable catalysts have commonly been explored from the modification method, introducing new material [99]. The different preparation methods and conditions of synthesis have brought several important catalyst key elements that influenced the biofuel quality. These key elements were the physicochemical properties of the catalysts such as the structure, surface area, porosity, stability, and bifunctional acid–base properties.

### 4.1. Catalyst Structure, Surface Area, and Porosity

The improvement of the dolomite catalyst’s physical properties greatly influenced the yield of the biofuel produced. Predominately, the dolomite catalyst modification greatly changed the structure, surface area, and porosity to allow reaction improvements in the catalytic reactions. As reported by Islam et al. [51], although the calcination process changed the material’s friability, the activated dolomite improved the catalytic activity due to the increases in the surface area and pore size. Moreover, the addition of metal to the activated dolomite catalyst clearly increased the porosity, which further improved the catalytic activity. As shown in Figure 9, the porosities of the fluffy-activated dolomite particles increased compared to the raw and activated dolomite.

In addition, the changes in the porosities of the catalysts also reduced the reaction temperatures and the coke formations during the catalytic reaction. As highlighted by Jumluck et al. [108], the high availability metal oxide particles and low coke formations at a 500 °C reaction temperature on the dolomites’ surfaces increased the catalytic activity involved in the reaction at low temperatures. This was further strengthened by Ruiz et al. [109], who considered that metal and oxide interfaces significantly provided oxygen vacancies through oxygen lost or transferring to the adsorbed species with changes on the surface composition by the reducibility of an oxide. In recent findings, Hafriz et al. [52] highlighted that the presence of metal on the activated dolomite catalyst altered and exerted a significant effect on the morphology of the catalyst, which transformed into a spherical structure from the initial cuboid shape. These certain characteristics of the modified dolomite catalyst included the high surface area and the structures of the catalysts that were favorable for the catalytic reaction activity. Figure 10 shows the TEM photomicrographs of the raw dolomite catalyst and the modified metal dolomite catalyst (PRE/Ni/CMD900) after radiation, with the electron and chemical compositions of the catalysts.

### 4.2. Catalyst Bifunctional Acid–Base Properties

The bifunctional properties of activated dolomite-based catalyst are another main significant key element for upgrading the quality of biofuel conversion. Naturally, dolomite catalysts have the basic properties of CaO–MgO alkaline earth metals. Nevertheless, the modifications on the activated dolomite catalyst’s surface enhanced the catalyst’s characteristic by imposing bifunctional acid–base properties. As highlighted by Shamsuddin et al. [110], the addition of 10% NiO/dolomite resulted in the strongest and highest acidic sites, referring to the highest intensity of an N–H bonding group. However, the decreased basic site densities (nCO_2_) of the catalyst showed the lowest CO_2_ desorption due to the acid strength from the NiO. The strong basic properties promoted the high CO_2_ molecules’ adsorption (acidic) on the catalyst’s plane. As reported by Hafriz et al. [61], these bifunctional acid–base properties further enhanced the catalytic reaction, which resulted in high producibility in the biofuel conversion. Figure 11 shows the highest amount of adsorption capacity for the dolomite catalyst that significantly proved the bifunctional acid–base properties of the modified NiO/dolomite catalyst.

### 4.3. Catalyst Activity and Selectivity

The catalyst’s activity and reaction behaviors are majorly dependent on the nature of the support used. Commonly, the strong activity of the dolomite catalyst is mostly generated through CaMg(CO_3_)_2_ transformation into active CaO and MgO, as well as the eliminations of CO_2_ by the temperature activation process (calcination). In addition, the activation of the dolomite catalyst could be further enhanced by reducing the coke deposition on the catalyst’s surface, eliminating the CO_2_ and preventing CaO carbonation [111].

At present, the dolomite catalyst’s modification has been explored extensively to improve and enhance the catalytic activity through the synthesis alteration of the catalyst as well as the optimization of the process. As studied by Hafriz et al. [52], compared to the co-precipitation and impregnation method, the precipitation technique had a greater impact on the catalytic activity of Ni/Dolomite catalysts due to the high surface areas and mesopores structures of the catalysts that were favorable for the catalytic reaction. In addition, the impact of the process optimization for the dolomite catalysts synthesized also influenced the catalytic activity. The excessive alteration of the active sites may have induced secondary reactions that generated coke formations and directly reduced catalytic activity [112]. Moreover, the optimum reaction temperature for the catalyst reaction also had a significant effect on the catalytic activity. As studied by Hafriz et al. [95], dolomite catalysts have indicated maximum biofuel conversion for catalytic reactions at optimum condition of 410 °C, 60 min, 5.50 wt% of catalyst loading, and 175 cm^3^/min of N_2_. The higher temperature could slow the cracking activity and induce major vaporization from the conversion reaction, which would reduce the biofuel producibility [86,101].

Other than that, the addition of a transition metal was the main promoter that increased the catalytic activity of the dolomite catalyst. As mentioned earlier, the addition of Fe, Zn, Cu, and Co in the activated dolomite catalyst synthesis played an important role in tuning the biofuel conversion yield towards a desired green biofuel [77]. The metals doped on the catalysts’ surfaces formed a wide channel for reactant diffusion into the catalysts’ pores, which enhanced catalytic activity [113]. This condition clearly enhanced the catalytic activity of the dolomite catalysts, which affected the performances of the catalytic reactions. The synergized contribution greatly influenced the enhancement of the dolomite catalytic activity in the reactions.

### 4.4. Catalyst Stability

Dolomite catalyst stability has been extensively explored in catalyst synthesis methods, modifications, and process optimizations. Thermal stability, chemical stability, and reaction stability are the significant properties that highly influence the behaviors of the dolomite catalytic reactions. Predominately, the high basicity of the dolomite catalyst has provided a wide support of particle dispersions for acid site attachment modifications. As reported by Shamsuddin et al. [110], the high dispersion of NiO particles on the dolomite catalyst contributed to maintaining the active sites in the catalyst’s support and increased catalytic stability. In addition, the dolomite catalyst’s thermal stability is also influenced by the metal doping modification and process optimization. The amount of metal loading on the dolomite catalyst’s surface significantly affects thermal stability. As explained by Kanchanatip et al. [82], higher MgCO_3_ loading led to lower thermal stability in the catalysts compared to lower MgCO_3_ loading. A lower thermal stability of a catalyst would affect the decomposition of the catalyst at an early stage of the catalytic reaction [114]. Meanwhile, the catalyst stability during the catalytic reaction was also contributed to by the amount of CO_2_ desorption during the reaction. As reported by Hafriz et al. [95], the amount of desorbed CO_2_ through the NiO-CMD catalyst could improve the stability of the catalyst during the reaction. Interestingly, the high desorption of CO2 indicated the strong interaction between the catalyst and the reactant, which significantly contributed to the dolomite catalyst. Thus, the catalyst stability affected the yield producibility in the biofuel conversion process.

## 5. Challenges and Future Direction

The progress of dolomite in catalytic pyrolysis has shown significant contributions for upgrading the quality of triglyceride biofuel conversions. At present, the dolomite catalyst has proven its effectiveness through the synthesis modifications on the catalyst’s surface, which produced the highest yield of pyrolytic products. In addition, the improvement of the process and the transition metal addition further increased the performance of the dolomite catalyst for the catalytic pyrolysis reaction. Nevertheless, there are several challenges in the dolomite catalyst that limit the application of the biofuel produced from the catalytic reactions. As a matter of fact, the acidity values of the biofuels produced are slightly higher compared to the ASTM standard, which requires biofuel to be blended with commercial fuels before further utilization. As reported, the high acid value will contribute to the large effect on the corrosion value, cold filter plugging point, and freezing point of the biofuel produced [101]. Although the dolomite catalyst has tremendously lowered acidity values and meets other properties of biofuel standards, complete fulfilment of the standards is required for biofuel to be commercialized. It is probable that the high acid value of the biofuel is due to excess of the bifunctional acid–base properties of the catalyst that leached to the biofuel. In addition, this also might be due to both the CaO carbonation and coke deposition on the reaction, which brought the dolomite deactivation that increased the acidity. The formation of hydrocarbons with acidic functional groups such as carboxylic acids and alcohols (phenols) could also be the reason for the produced acid values of the biofuel [92]. Therefore, further investigation should be focused on catalyst synthesis modification, transition metal selection, and the factors to further reduce the acidity values. The interaction of the catalyst bonding to the reactants and its chemical composition should be more emphasized in knowing the roots of the high acidity of the produced biofuel.

On the other side, one of the main challenges is the deactivation of the catalyst over time due to the accumulation of coke and other carbon-based materials. This can lead to a decrease in the activity of the catalyst and an increase in the amount of energy required to maintain the reaction. Another challenge is the low selectivity of the dolomite catalysts, which can lead to the production of a wide range of products, including undesired by-products such as tar and char. In the future, research is needed to overcome these challenges and improve the performances of dolomite catalysts in catalytic pyrolysis reactions. This can include the development of new catalyst formulations, the use of advanced characterization techniques to better understand the behavior of the catalyst, and the optimization of reaction conditions to improve selectivity and activity.

Additionally, soaps are a common by product of catalytic pyrolysis reactions, and they can be formed when fatty acids present in the feedstock react with the dolomite catalysts. Soaps can also form due to the reaction of fatty acids with water, which is often present in the feedstock. However, the formation of soaps can reduce the yield of bio-oil produced in catalytic pyrolysis reactions. Soaps can form a complex mixture of products that are difficult to separate from the bio-oil, and they can also cause the bio-oil to become more viscous, making it more difficult to handle and transport. In addition, soaps can also accumulate in the reactor, which will reduce the catalyst activity and lead to the deactivation of the catalyst over time. At present, the modification of acid sites on the surface of the dolomite catalysts has helped to promote the cracking of triglycerides and reduce the formation of soaps. Other methods include using drying procedures to remove the water from the feedstock and using the catalysts at higher temperatures to promote the cracking of triglycerides and reduce the formation of soaps. Nevertheless, there are plenty of studies about the effect of the saponification process using dolomite catalysts in catalytic pyrolysis reactions.

Recent advances in triglyceride catalytic pyrolysis via heterogeneous dolomite catalysts have the potential to become more applicative strategies and contribute to the next step of implementation by addressing some of the challenges and limitations associated with biofuel production. Further research and development can focus on optimizing the dolomite catalysts, including their composition, structure, and preparation methods, to improve their catalytic activity, stability, and selectivity towards desired biofuel products. This may involve exploring different dolomite sources, modifying catalyst properties through doping, impregnation, or other techniques, and optimizing catalyst preparation and activation methods. Critically, choosing the right catalyst with the appropriate properties, such as activity, selectivity, and stability, for the specific pyrolysis conditions and desired product outcomes, can be challenging. Identifying catalysts that can effectively promote the desired chemical reactions while minimizing undesirable side reactions is critical. Catalysts can deactivate over time due to fouling, coking, or poisoning, reducing their effectiveness and requiring regeneration or replacement. Catalyst deactivation can impact the yield and quality of biofuel products and can pose challenges in maintaining stable catalytic performance over prolonged pyrolysis runs. Optimizing the pyrolysis conditions, such as temperature, pressure, and residence time, to achieve high yields of desired biofuels while minimizing the formation of byproducts, can be challenging. Finding the right balance between achieving the high conversion of triglycerides and minimizing undesired reactions that lead to byproduct formation can be complex. In addition, the composition and properties of the feedstock, such as triglyceride type, moisture content, and impurities, can vary, which can impact the performance of the catalytic pyrolysis process. Feedstock variability can affect the conversion, product yield, and product distribution, posing challenges in achieving consistent and high-quality biofuel products.

Moreover, the quality and stability of the biofuel products, such as their heating value, viscosity, stability, and storage characteristics, are critical for their performance and marketability. Ensuring consistent and high-quality biofuel products with desirable properties can be challenging, as the pyrolysis process and catalyst performance can influence the final product’s characteristics. In addition, the integration of triglyceride catalytic pyrolysis with other biofuel production technologies, such as hydro processing, bio-oil upgrading, or biorefinery concepts, can create synergies and improve the overall economics and sustainability of the process. This may involve developing integrated process configurations, optimizing process interfaces, and evaluating the benefits of process integrations in terms of product quality, yield, and overall process performance. Overall, the applicability of recent advances in triglyceride catalytic pyrolysis via heterogeneous dolomite catalysts can be enhanced by addressing the challenges and exploring opportunities for optimization, diversification, integration, and stakeholder engagement in the biofuel production landscape.

## 6. Conclusions

In conclusion, the recent advances in triglyceride catalytic pyrolysis using heterogenous dolomite catalysts for upgrading biofuel quality have shown great potential to produce renewable and sustainable fuels. The use of dolomite catalysts has been found to be effective in catalyzing the pyrolysis of triglycerides to produce high-grade biofuels. Dolomite has become a popular catalyst due to its affordability and abundance for tar removal. Calcined dolomites, with their large internal surface areas and high oxide contents, are more active than uncalcined dolomites. The presence of CaO is important for dolomite’s effectiveness. However, the interaction of CaO and MgO and the presence of iron compounds also play a role. The in situ and ex situ use of dolomite is effective compared to other catalysts such as sand, FCC, olivine, and γ-alumina. Dolomite and calcined dolomite can be combined with other catalysts such as nickel and Fe_2_O_3_. The modification of the dolomite catalyst’s surface with metal/metal oxide nanoparticles and the optimization of the synthesis process have further improved the catalytic activity and selectivity of the process. The recent investigation has extensively focused on the physicochemical properties of the catalysts to improve their catalytic performance, including their structures, surface areas, porosities, stabilities, and bifunctional acid–base properties. However, challenges such as maintaining catalytic stability and addressing the slightly higher acid value of the biofuels produced compared to the ASTM standard must be addressed to ensure commercial viability. Overall, the recent advances in triglyceride catalytic pyrolysis using heterogenous dolomite catalysts offer a promising avenue for producing sustainable and high-quality biofuels, contributing to a greener and more sustainable future.

## Figures and Tables

**Figure 1 nanomaterials-13-01947-f001:**
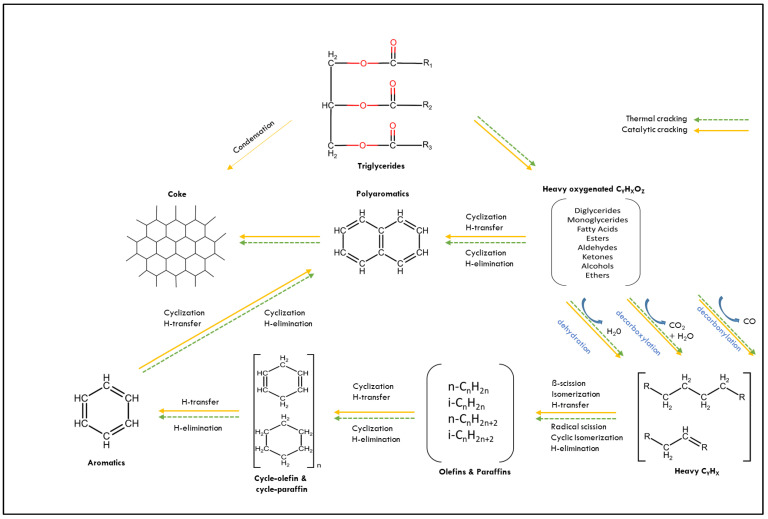
Schematic diagram of catalytic cracking reaction of triglycerides.

**Figure 2 nanomaterials-13-01947-f002:**
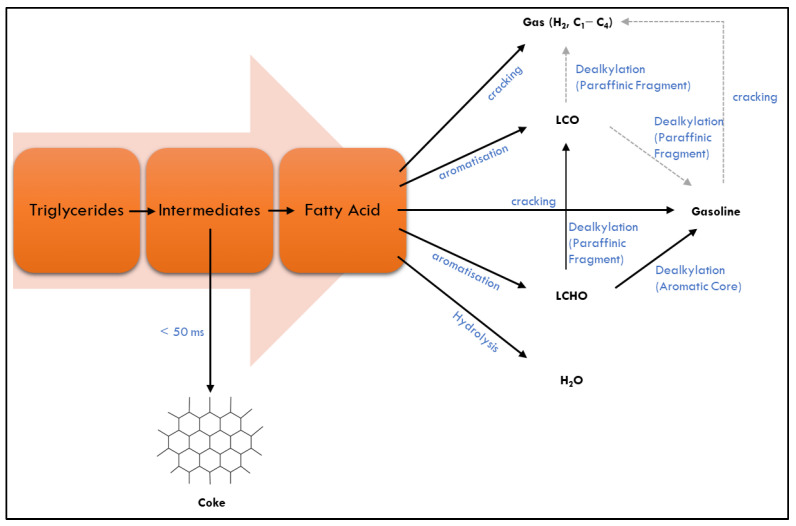
Triglycerides catalytic conversion into various biofuel products.

**Figure 3 nanomaterials-13-01947-f003:**
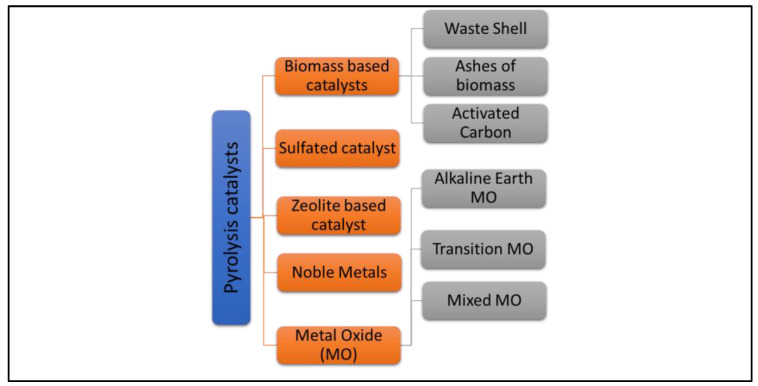
The types of catalyst groups in catalytic pyrolysis for biofuel production.

**Figure 4 nanomaterials-13-01947-f004:**
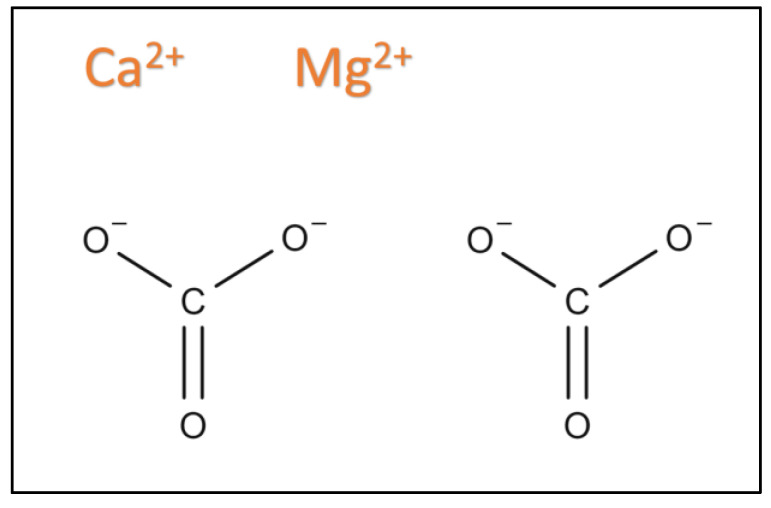
Dolomite Chemical Structure.

**Figure 5 nanomaterials-13-01947-f005:**
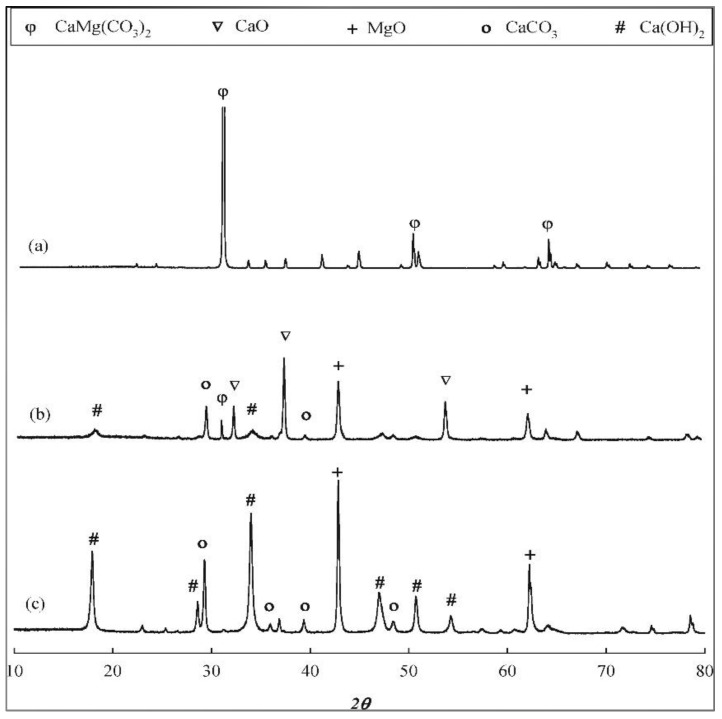
XRD (**a**) raw, (**b**) activated and (**c**) post reaction of dolomite catalyst Reprinted with permission from Ref. [93]. 2018, Jinsoo Kim.

**Figure 6 nanomaterials-13-01947-f006:**
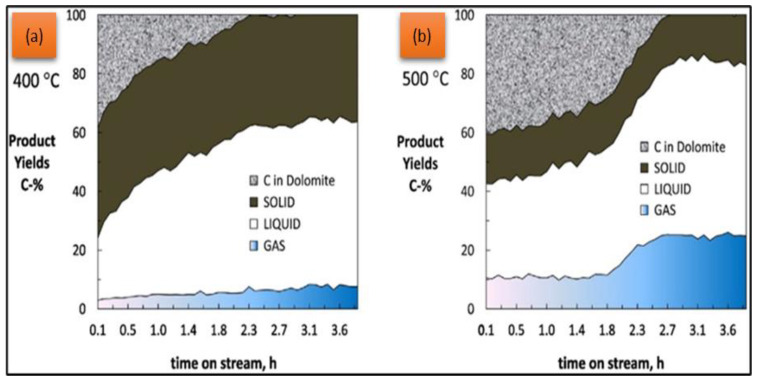
Evolution with time on stream of carbon yields (C wt%) in dolomite, solid (pyrolytic lignin), liquid (upgraded bio-oil) and gas products. Reaction temperature: (**a**) 400 °C and (**b**) 500 °C Reprinted with permission from Ref. [92]. 2019, Beatriz Valle.

**Figure 7 nanomaterials-13-01947-f007:**
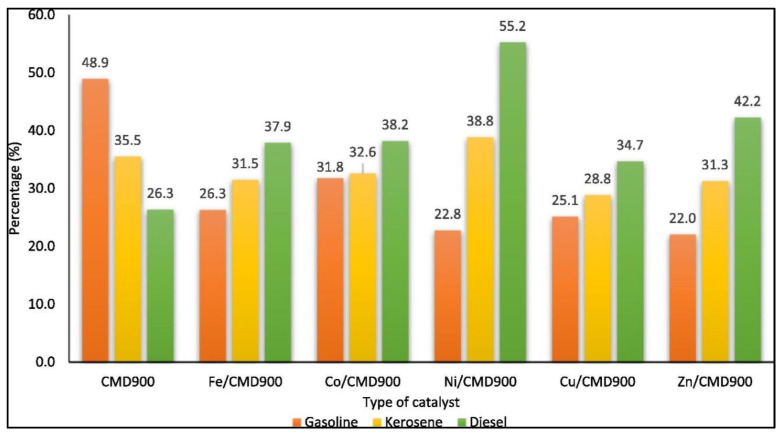
Comparison of transition-metal-doped activated dolomite catalysts with compositions of biofuel in the produced pyrolysis oil Reprinted with permission from Ref. [77]. 2020, Hafriz R.S.R.M.

**Figure 8 nanomaterials-13-01947-f008:**
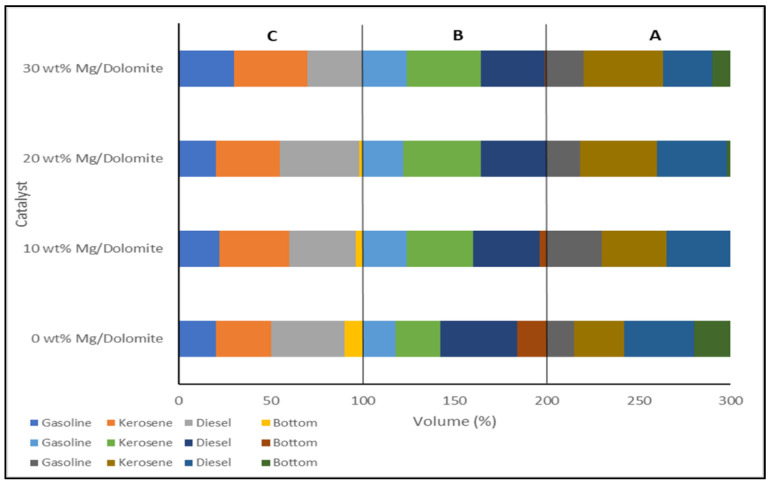
The biofuel improvement of Mg/dolomite catalyst at (**A**) 400 °C, (**B**) 450 °C and (**C**) 500 °C Adapted with permission from Ref. [82]. 2022, Kitirote Wantala.

**Figure 9 nanomaterials-13-01947-f009:**
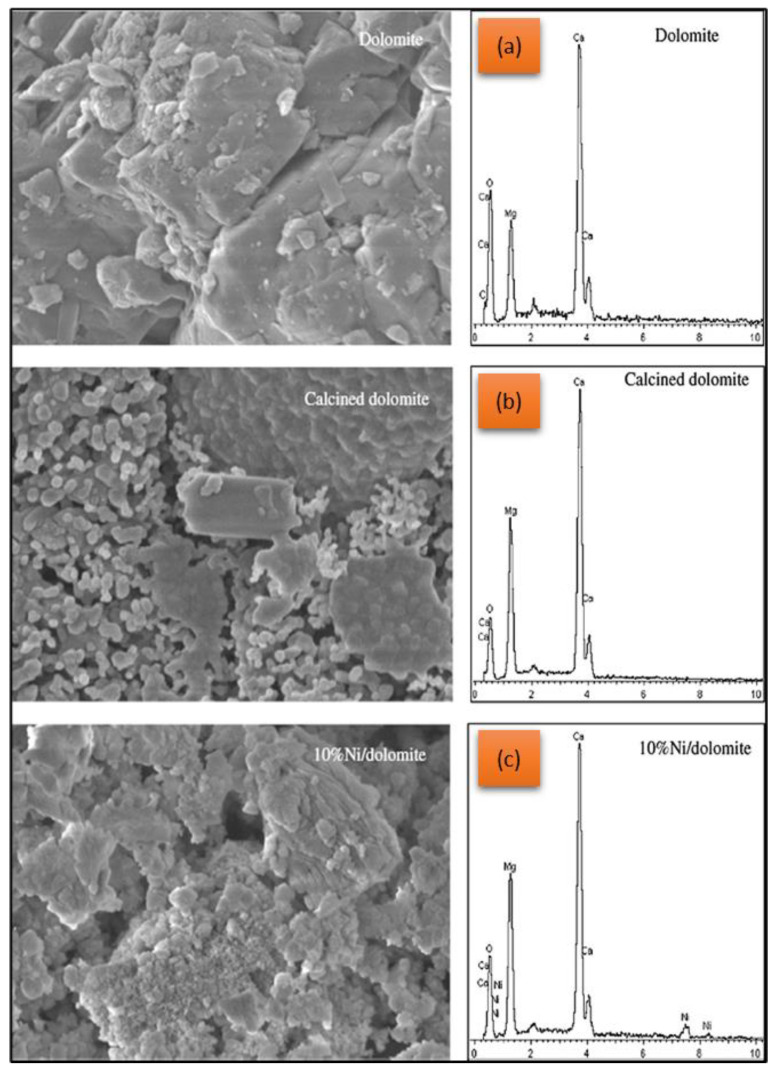
The SEM/EDX comparison of (**a**) raw, (**b**) calcined, and (**c**) Ni of dolomite catalyst. Reprinted with permission from Ref. [51]. 2020, Islam M.W.

**Figure 10 nanomaterials-13-01947-f010:**
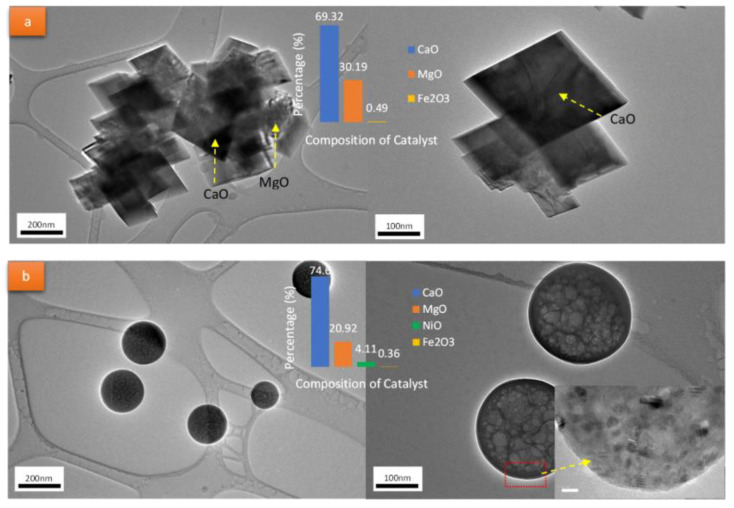
TEM photomicrographs of (**a**) raw dolomite catalyst and (**b**) modified metal dolomite catalyst (PRE/Ni/CMD900) [52]. Reprinted with permission from Ref. [52]. 2021, Hafriz, R.S.R.M.

**Figure 11 nanomaterials-13-01947-f011:**
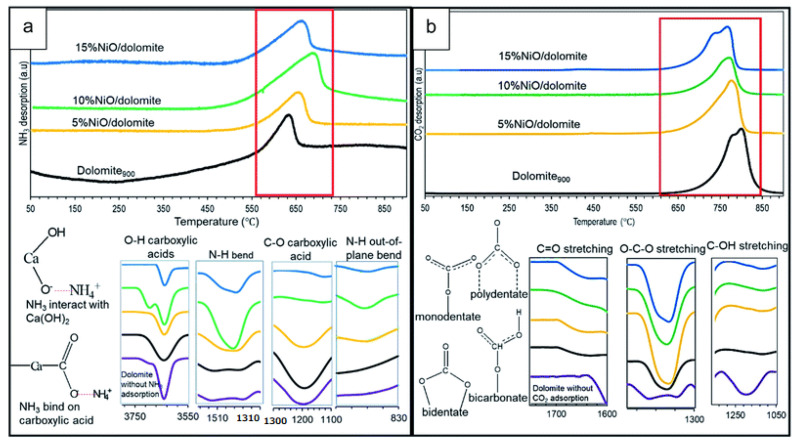
Bifunctional acid–based properties for (**a**) catalyst desorption and NH_3_ absorbed and (**b**) catalyst desorption and CO_2_ adsorbed Reprinted with permission from Ref. [110]. 2021, Taufiq Yap.

**Table 2 nanomaterials-13-01947-t002:** The chemical and physical properties of dolomite catalysts [51].

Chemical Formula	Key Constituents	Trace Elements	Color	Specific Gravity
CaMg(CO_3_)_2_	CaO, MgO and CO_2_	SiO_2_, Fe_2_O_3_ and Al_2_O_3_	White, Pink, Green, Brown, and Black	2.8 to 2.9 g/cm^3^

**Table 3 nanomaterials-13-01947-t003:** The composition of natural dolomite catalysts [85,86].

Elemental Analysis (wt%)	Sweden Dolomite	Thailand Dolomite	China Dolomite	Malaysia Dolomite
CaO	30.50	65.30	30.72	38.97
MgO	20.20	34.60	20.12	39.79
Al_2_O_3_	0.11	-	0.14	0.16
Fe_2_O_3_	0.54	623 ppm	0.03	0.08
SiO_2_	2.21	-	2.01	0.098
K_2_O	0.04	-	0.02	-
MnO	0.06	-	0.002	-

**Table 4 nanomaterials-13-01947-t004:** The recent work of dolomite catalysts for catalytic pyrolysis reactions.

Feedstock	Catalyst Type	Process Condition	Findings	Reference
Leucaena leucocephala Oil	Calcined dolomite and zeolite catalyst	- Catalysts were calcined at 500 °C (4 h) 500 g of the catalyst feed into reactor. - N_2_ fluidized gas purged at a flow rate of 20 L min^−1^ - Feedstock pyrolyzed in the fluidized bed reactor	Dolomite facilitated the cracking reaction and increased the light bio-oil yield. The viscosities of the bio-oil products obtained using the catalysts were lower by 60% of that of the bio-oil produced without dolomite or zeolite.	[91]
Pine sawdust Bio-Oil	Calcined-activated dolomite catalyst	- Flash pyrolysis with 5 t/h capacity in a conical rotary reactor. - Pyrolyzed at 400 °C and 500 °C for 4 h.	48% high quality biofuel conversion: Ketones (37%) Cyclo pentanones (30%)	[92]
Liriodendron Oil	Calcined-activated dolomite catalyst	- Pyrolysis temperature, ranging from 400 °C to 550 °C - Feeding rate of 100 g/h - N_2_ gas purged flow rate of 7.5 L min^−1^	Dolomite HHVs (23.09–28.02 MJ/kg) higher than those of bio-oil from the sand (21.64–24.37 MJ/kg). High H_2_/CO ratio in the gas product, which can be used in the synthesis of liquid fuel	[93]
Saccharum officinarum LOil	Calcined dolomite	- Pyrolysis temperature 450 °C - N_2_ sweep gas 80 cm^3^ min^−1^, - 10% catalyst feeding	Calcined dolomite upgraded bio-oil with a lower oxygen content, higher gross calorific value, and decreased acid corrosion	[94]
Waste Cooking Oil	Ni-doped-calcined Malaysiadolomite (Ni/CMD900) catalyst	- Deoxygenation reaction time 30 min - Process temperature 390 °C - Reaction mixture 150 g of WCO and 5 wt% of synthesized catalysts in 1000 mL reactor.	PRE/Ni/CMD900 catalyst resulted superior deoxygenation reaction activity with high conversion of WCO (68.0%), high yield of pyrolysis oil (36.4%), and less coke formation (32.0%)	[52]
Waste Cooking Oil	Ni, Fe, Zn, Cu, Co/CMD900	- 150.0 g of WCO and 7.5 g of catalyst added to the reactor - N_2_ flow rate 150 cm^3^/min - Reaction temperature of 390 °C for 30 min	Ni/CMD900 catalyst exhibited highest conversion (67.0%) and high selectivity (80.2%) with high proportion of saturated linear hydrocarbons	[77]
Waste Cooking Oil	Dolomite added with magnesium.carbonate (MgCO_3_) (0–30 wt%),	- 500 °C reactions temperature - 20 wt% Mg/dolomite catalyst	The highest production yield 84 vol% Light biofuels yield 65 vol%.	[82]
Waste Cooking Oil	NiO-dolomite catalyst	- 410 °C reaction temperature - 5.50 wt% of catalyst 175 cm^3^/min of N^2^ gas flow rate.	The bio-oil meets the requirements of diesel fuel. The biofuel characterization is tested with several standard parameters	[95]

**Table 5 nanomaterials-13-01947-t005:** Comparison of the biofuel properties using dolomite-based catalysts with ASTM.

Catalyst	Heating Value	Acid Value	Reference
	ASTM 44–45 (MJ/Kg)	ASTM <0.01 mg (KOH/g)	
Activated Dolomite	Not Available	33	[101]
Mg/Activated Dolomite	42.14	2.81	[82]
Ni/Activated Dolomite (Precipitation)	Not Available	47	[52]
Ni/Activated Dolomite(Co-Precipitation)	Not Available	49	[52]
Ni/Activated Dolomite (Impregnation)	Not Available	58	[52]
NiO/Activated Dolomite	43.83	26.6	[95]
Fe/Activated Dolomite	Not Available	40	[77]
Co/Activated Dolomite	Not Available	64	[77]
Cu/Activated Dolomite	Not Available	75	[77]
Zn/Activated Dolomite	Not Available	78	[77]

## Data Availability

Not Applicable.

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
