# Peer review of "Recent Advances of Triglyceride Catalytic Pyrolysis via Heterogenous Dolomite Catalyst for Upgrading Biofuel Quality: A Review"

_nanomaterials, 2023, doi:10.3390/nano13131947_

Round 1

Reviewer 1 Report

The authors should add a new idea to this type of review study. The novelty and potential impact of this paper should be clarified. I recommend that you edit the entire article according to journal standards and re-send it. An article that is not well prepared is more of a research technical report, They should be able to propose a model to cover a gap based on the literature and they should evaluate the model based on justifications and comparisons concerning those proposed in the literature. Finally, the readers of this paper should be able to get the main points through some results and summary.
- Most of the ideas written were already described in many kinds of literature. The Authors tried to compile it but lack of the enhancement of the interrelation analysis between the references. It is advised that the authors give a deeper analysis of how these ideas become more applicative strategies so that they can contribute to the next step of implementation.
- Novelty: The authors partly stated what motivated the idea portrayed in this study, what then does this study offer beyond the recent advances made on the combined approaches? I would advise the authors to carefully carry out a close comparison as well in the discussion section to enumerate the advantages this study offers over other related works
- Introduction provides a great overview and introduces the topic. However, it misses the aim of the study or what is going to be done in this study. Please state clearly the aims and what this study does. However, highlights that innovative technologies combined with renewable fuels are able to improve engine emissions and fuel economy ( 10.1016/j.pecs.2006.08.003, (10.1080/15567036.2022.2124326, 10.1007/978-981-16-8751-8_3, /10.1016/j.renene.2005.12.003). The authors could extend the introduction discussion reporting that innovative technologies could give a potential boost to the engine fuel economy and engine-out emissions reduction.
- More in-depth analysis of the author's contribution to this paper in the introduction section. I would like to see more discussion of the literature so that I can clearly identify the article relates to competing ideas.
- The language of the manuscript is fair; I would advise consulting a language editor to further polish the language of the manuscript. There are several grammatical mistakes.
- Challenges and future directions to improve and implement of these technologies with big data analytics should be discussed.
- In my opinion, there are several up-to-date approaches to the idea. Authors should look at these approaches, compare the results and prove their idea. This is a major concern.
- The conclusions don't tie to the discussion well and should be reconsidered. There needs to be a clearer discussion of the points in the body, or the conclusions should be adjusted to better match the existing discussion.

Author Response

Dear Dr,

We would like to express our sincere appreciation for the time and effort that you spent on handling our manuscript. We have been indebted to you and all the reviewers for giving us the opportunity to improve our manuscript. We have done a series of amendments and did our best to revise the manuscript according to the comments from the reviewers. We strongly believe the manuscript is suitable to be published in Nanomaterials: Energy and Catalysis since it is within the scope covered by the journal. Responses to the comments are attached.

Reviewer 2 Report

Before publication, the following items should be addressed by the author:

1.       English must be improved.

2.       The novelty of the paper and the new approach to the topic should be highlighted.

3.       The problem statement should be developed.

4.       The Journal’s standards for referencing must be considered.

5.       A list of Abbreviations must be given.

6.       To indicate the importance of renewable energy and its role in the future, the following papers are suggested to be considered:

"Transition away from fossil fuels toward renewables: lessons from Russia-Ukraine crisis." Future Energy 1.1 (2022). https://doi.org/10.55670/fpll.fuen.1.1.8

"An overview of renewable energy technologies for the simultaneous production of high-performance power and heat." Future Energy 2.2 (2023): 1-11. https://doi.org/10.55670/fpll.fuen.2.2.1

7.       Clarify the main factors that affect the conversion, product yield, and product distribution in biofuel production using pyrolysis.

8.       Explain how catalysts influence the mechanisms of catalytic pyrolysis of triglycerides and the formation of desired products?

9.       What are the challenges in optimizing the catalytic pyrolysis process to produce high yields of desired biofuels with minimum byproducts?

10.   Can alternative feedstocks be utilized in catalytic pyrolysis to produce renewable bio-based products for fuel applications?

11.   How does the chemical composition of dolomite catalyst contribute to its catalytic activity in bio-oil upgrading?

12.   What are the challenges associated with the use of raw dolomite catalyst in bio-oil upgrading and how can they be addressed?

13.   Explain the potential of CaO as an alternative deoxygenation catalyst for biofuel production and how does it compare to other catalysts such as ZnO?

14.   Highlight the most effective methods for synthesizing and modifying dolomite-based catalysts for biofuel conversion.

15.   Explain the most important factors affecting the stability of dolomite catalysts and how can they be optimized to maintain high catalytic activity over time?

Author Response

(The authors gave the same response as above.)

Reviewer 3 Report

The subject of the paper clearly falls within the scope of this Journal.

The paper is very interesting, well written and well organized, and represents some advancement over the actual state-of-the-art. The ways and means are well described as well as the obtained results which are thoroughly discussed and conclusions are well drawn. The paper is also supported by relevant literature. However, in order to make for a stronger paper, I suggest that the authors should cite and discuss the following relevant papers, which could be used as benchmark towards the proposed approach:

– Catarino et al., “Calcium diglyceroxide as a catalyst for biodiesel production”, Journal of Environmental Chemical Engineering, 7(3): 103099 (2019) - DOI: 10.1016/j.jece.2019.103099

- Dias et al., “Review on biodiesel production processes and sustainable raw materials”, Energies, 12, 4408 (2019) - DOI: 10.3390/en12234408

  I do recommend the publication of this paper, subjected to these changes.

Author Response

(The authors gave the same response as above.)

Round 2

Reviewer 1 Report

Ok

Reviewer 2 Report

Accept

Reviewer 3 Report

The authors have reviewed.